# Gynecologic Cancer Risk and Genetics: Informing an Ideal Model of Gynecologic Cancer Prevention

Lauren C. Tindale [1], Almira Zhantuyakova [1], Stephanie Lam [1], Michelle Woo [1], Janice S. Kwon [1], Gillian E. Hanley [1], Bartha Knoppers [2], Kasmintan A. Schrader [3,4], Stuart J. Peacock [5], Aline Talhouk [1], Trevor Dummer [6], Kelly Metcalfe [7,8], Nora Pashayan [9], William D. Foulkes [10], Ranjit Manchanda [11,12,13], David Huntsman [14], Gavin Stuart [1], Jacques Simard [15] and Lesa Dawson [1,*]

1    Department of Obstetrics and Gynaecology, University of British Columbia, Vancouver, BC V6T 1Z4, Canada; lauren.tindale@ubc.ca (L.C.T.); azhantuyakova@bccrc.ca (A.Z.); stlam@bccrc.ca (S.L.); miwoo@bccrc.ca (M.W.); janice.kwon@vch.ca (J.S.K.); gillian.hanley@vch.ca (G.E.H.); a.talhouk@ubc.ca (A.T.); gavin.stuart@ubc.ca (G.S.)
2    Faculty of Medicine, Human Genetics, Centre of Genomics and Policy, McGill University, Montreal, QC H3A 0G4, Canada; bartha.knoppers@mcgill.ca
3    Hereditary Cancer Program, BC Cancer, Vancouver, BC V5Z 4E6, Canada; ischrader@bccancer.bc.ca
4    Department of Medical Genetics, University of British Columbia, Vancouver, BC V6T 1Z4, Canada
5    Faculty of Health Sciences, Simon Fraser University, Vancouver, BC V5A 1S6, Canada; speacock@bccancer.bc.ca
6    School of Population and Public Health, University of British Columbia, Vancouver, BC V6T 1Z4, Canada; trevor.dummer@ubc.ca
7    Women's College Research Institute, Toronto, ON M5G 1N8, Canada; kelly.metcalfe@utoronto.ca
8    Lawrence S. Bloomberg Faculty of Nursing, University of Toronto, Toronto, ON M5S 1A1, Canada
9    Department of Applied Health Research, University College London, London WC1E 6BT, UK; n.pashayan@ucl.ac.uk
10   Departments of Medicine, Human Genetics and Oncology, McGill University, Montreal, QC H3A 0G4, Canada; william.foulkes@mcgill.ca
11   Wolfson Institute for Population Health, CRUK Barts Cancer Centre, Queen Mary University of London, Charterhouse Square, London EC1M 6BQ, UK; r.manchanda@qmul.ac.uk
12   Department of Health Services Research and Policy, London School of Hygiene & Tropical Medicine, London WC1H 9SH, UK
13   MRC Clinical Trials Unit at UCL, Institute of Clinical Trials & Methodology, Faculty of Population Health Sciences, University College London, London WC1E 6BT, UK
14   Department of Pathology and Laboratory Medicine, University of British Columbia, Vancouver, BC V6T 1Z4, Canada; dhuntsma@bccancer.bc.ca
15   Genomics Center, Centre Hospitalier Universitaire de Québec-Université Laval Research Center, Québec City, QC G1V 4G2, Canada; jacques.simard@crchudequebec.ulaval.ca
*    Correspondence: lesa.dawson@vch.ca

**Abstract:** Individuals with proven hereditary cancer syndrome (HCS) such as *BRCA1* and *BRCA2* have elevated rates of ovarian, breast, and other cancers. If these high-risk people can be identified before a cancer is diagnosed, risk-reducing interventions are highly effective and can be lifesaving. Despite this evidence, the vast majority of Canadians with HCS are unaware of their risk. In response to this unmet opportunity for prevention, the British Columbia Gynecologic Cancer Initiative convened a research summit "Gynecologic Cancer Prevention: Thinking Big, Thinking Differently" in Vancouver, Canada on 26 November 2021. The aim of the conference was to explore how hereditary cancer prevention via population-based genetic testing could decrease morbidity and mortality from gynecologic cancer. The summit invited local, national, and international experts to (1) discuss how genetic testing could be more broadly implemented in a Canadian system, (2) identify key research priorities in this topic and (3) outline the core essential elements required for such a program to be successful. This report summarizes the findings from this research summit, describes the current state of hereditary genetic programs in Canada, and outlines incremental steps that can be taken to improve prevention for high-risk Canadians now while developing an organized population-based hereditary cancer strategy.

**Keywords:** cancer screening; hereditary cancer syndrome; population-based genetic testing; BRCA

## 1. Introduction

A significant number of gynecologic cancers in Canada can be prevented. Beyond risk factors that can be modified by lifestyle or HPV vaccination, a meaningful component of an individual's cancer risk can be attributed to hereditary predisposition or genetic susceptibility.

Individuals with proven hereditary cancer syndrome (HCS) have elevated lifetime rates of malignancy. It is estimated that one-fifth of ovarian cancers are hereditary in origin, with *BRCA1* and *BRCA2* genes being the largest contributor. Other HCS genes associated with increased ovarian cancer risk include Lynch Syndrome mismatch repair genes (*MMR*), *TP53, RAD51C, RAD51D, BRIP1*, and *PALB2* [1]. Women with *BRCA1-* or *BRCA2*-associated HCS have a lifetime risk of ovarian cancer between 17–44%. HCS genes associated with endometrial cancer include Lynch syndrome (*MMR*) and *PTEN* pathogenic variants. Cervical cancer is not considered a hereditary cancer, although mutations in immune response-related genes may be related to the ability to clear human papillomavirus infections [2]. Vulvar cancer is related to human papillomavirus infection as well as *TP53* [3]. The full landscape of genetic variation in gynecological cancers has yet to be fully elucidated.

Advances in the fields of genomics, genetics, and hereditary cancer prevention offer new opportunities for women to quantify cancer risk and access effective prevention. It is now known, for example, that women with *BRCA1* mutations have dramatically elevated risks of cancer (including ovarian (44%) and breast (71%) cancers [4]) and that high-risk surveillance and risk-reducing salpingo-oophorectomy (RRSO) decrease mortality by as much as 70% [4]. It is important to note that effective screening still does not exist for ovarian cancer, and RRSO is the most effective prevention. Better identification of high-risk individuals and directed preventive care will decrease gynecologic incidence and mortality and help to build a more sustainable healthcare system.

To find carriers of *BRCA* only after a cancer diagnosis should be considered a failure of cancer prevention. Optimally, effective health systems would identify unaffected women with *BRCA* or other HCS, offering prophylactic interventions before the onset of disease. Although personalized prevention has been lifesaving in the management of individual cases one patient at a time, the promise of genetics and genomics has not yet been delivered in the domain of public/population health. There is a need to reconcile the disparity between knowledge and health policy. New and more effective models of care could decrease the burden, morbidity, and mortality of hereditary cancer and expand system capacity to provide this preventative health care to Canadians in an equitable way.

The long-established model of cancer genetics care in Canada requires that (1) patients undergo an individual session with a genetic counsellor before testing is offered, and (2) publicly funded tests are provided only to people with a proven cancer diagnosis or a significant family history. Not only does this model miss more than 50% of mutation carriers [5,6], but it has led to wait times that are unmanageable to the system and unacceptable to the public. Importantly, many marginalized groups in Canada experience barriers in accessing equitable and timely genetic services. How can health systems expand access to testing and prevention, and could testing be delivered more broadly at a population-based level?

For the purposes of this manuscript, population-based testing refers to a process which offers genetic testing for hereditary cancer predisposition to asymptomatic groups of persons in a jurisdiction, irrespective of cancer diagnosis or family history.

## 1.1. Current System of Hereditary Cancer Prevention

Despite the proven efficacy of personalized hereditary cancer prevention, most individuals with HCS are unaware that they are at elevated risk and do not access specialized prevention or screening. Large studies have reported population prevalence of HCS between 0.64–2.40% of the general unselected population [7–9]. It is estimated that fewer than 5% of those with *BRCA* pathogenic variants are being identified by the current model of family-based risk assessment and testing [10]. Although embedded oncology clinic-based testing [11,12] may improve these rates, it is clear that many opportunities for prevention are missed.

In most jurisdictions, healthy people without cancer may access testing only if a proven HCS mutation has been already found in a relative; however, the uptake of this family cascade testing is inconsistent. Many relatives do not present for assessment despite detection of HCS in the family or may not have been informed of or understood the implications of their HCS family history [13–18]. There is significant under-representation of certain groups in considering equitable access, both in testing and prevention that may be related to for those disadvantaged by socioeconomic status, education, or systemic racism.

Raised public awareness about hereditary cancer has led to dramatic increases in referrals for risk assessment across Canada, resulting in higher demand that has exceeded the capacity of the current model. In some centers, a woman with a cancer diagnosed at an early age may wait more than 18–24 months for a hereditary cancer assessment. Even if eligible, there are inequities amongst the patient populations accessing genetic testing. In the province of British Columbia, Canada, patients of white European ancestry are overrepresented in all referrals and testing, yet women of Asian ancestry are underrepresented; among Asians who were tested, however, there is a higher incidence of HCS mutations [19]. The same study found that Indigenous peoples represent a small fraction of all referrals, suggesting that there are either significant impediments or preferences in place preventing these women from accessing knowledge, personal power, medical advice, dedicated referral, the offer of testing, and the benefit of prevention. For many Indigenous people, the expectation of a detailed family history is a powerful barrier to care. These findings of underrepresentation in Asian and Indigenous populations were made based on comparison with population proportions represented in the 2016 Canadian Census [19].

Given the proven efficacy of preventative interventions in families with HCS and the limitations of the current model of care, health systems must now re-assess hereditary cancer prevention strategies and policy.

## 1.2. Combined Epidemiological Risk Factors, Pathogenic Variants, and Polygenic Risk Scores

The main goal of preventative programs is to find a balance between maximizing the population benefit while minimizing the harms: overtreatment, patient anxiety, unnecessary costs [20,21]. A tailored risk-based prevention program would estimate an individual's absolute risk of developing cancer and target those at high-risk who are most likely to benefit from prophylactic surgery or other prevention strategies [22]. A pilot study of a general population women who self-referred themselves for the study based on leaflets made available at primary care practices, demonstrated feasibility, acceptability, high uptake, and satisfaction with risk assessment approaches along with a reduction in cancer risk perception and worry with time [23]. General population surveys have shown that women are keen to know their personalized cancer risk and would undergo surgical prevention if found to be at increased risk [24,25].

Even before germline testing for HCS, an individual woman's risk of gynecologic cancer can be estimated considering epidemiologic, family, and genomic predictors. Although age thresholds are typically used for screening in most cancers, in the case of ovarian cancer, routine ultrasound screening is ineffective for early detection, and routine imaging is not recommended regardless of age or risk. The incidence of ovarian cancer is 1.3% in the general unselected population, and traditionally an estimated 10% lifetime risk has previously been considered the threshold for offering preventative oophorectomy. However,

recently there have been calls for broadening access to surgical prevention, with preventive oophorectomy at the 4–5% lifetime ovarian cancer risk threshold being shown to save 7–10 life years and found to be cost-effective [26]. Clinical practice in several international centers has changed with RRSO being offered at 5% lifetime risk. This is now supported by a Scientific Impact Paper from the Royal College of Obstetricians and Gynecologists [27] and a recent consensus meeting of the UK Cancer Genetics Group [28]. Of note, a diagnosis of ovarian cancer in a first-degree relative will alone raise an individual woman's estimated lifetime risk to 4–5%.

While pathogenic mutations in genes such as *BRCA1* and *BRCA2* confer high risks of breast and ovarian cancers, these account for only a small proportion of all cancer cases in the general population. Multifactorial risk assessment that considers other epidemiological factors, including menopausal status, BMI, and history of endometriosis, add value in understanding a person's true lifetime risk for ovarian cancer [29,30]. For all women, oral contraceptive use is highly protective in the prevention of ovarian/endometrial cancers, conferring >50% reduction in cancer after 5 years of use. Tubal ligation reduces ovarian cancer risk, and new research has confirmed that opportunistic salpingectomy in a healthy low-risk general population women at the time of elective hysterectomy or sterilization significantly reduces the risk of ovarian cancer—no serous ovarian cancers were observed in their group of over 25,000 individuals who underwent opportunistic salpingectomy, compared to 15 in the control group of 32,000—with researchers conservatively estimating potential prevention of at least 80% from this procedure [31]. RRSO remains the standard of care for *BRCA* carriers, although an early salpingectomy and delayed oophorectomy approach is currently being investigated in clinical trials [32,33].

Genomics can play a valuable role in risk prediction, and as technology progresses and sequencing costs reduce, testing has advanced far beyond the historic single-gene Sanger testing. Genome-wide association studies (GWAS) have identified variants that are common in the population and predict susceptibility. Multiple common breast cancer susceptibility variants discovered through GWAS, for instance, confer minimal risk individually, but their combined effect, summarized as a polygenic risk score (PRS), can be substantial; explaining up to 40% of risk [34–36]. Although PRS evidence in breast cancer differs when compared to ovarian cancer because of baseline disease prevalence, PRS has been validated to estimate ovarian risk and can be used as a framework for research and prevention in gynecologic cancers [37]. As these factors combine in a multiplicative rather than an additive fashion [38], integrating genomic factors with clinical practice could accelerate the translation of innovation into care.

Optimally, gynecologic cancer prevention would provide every woman with risk assessment incorporating epidemiologic and family history factors in parallel with germline genetic testing/genomic risk prediction such that prevention is directly focused on those women who stand to benefit the most.

### 1.3. Evidence from Population-Based Genetic Testing Trials

Studies implementing versions of genetic population-based testing in various countries have demonstrated that wide-scale implementation can provide cancer prevention without adversely affecting participants (select studies listed in Table 1). Population-based *BRCA* testing can identify carriers of HCS in individuals with previous cancer diagnosis [39], those of Ashkenazi Jewish descent [7,40–42], and in the general population [8,9,43,44]. While *BRCA1* and *BRCA2* are the most well-studied genes, inclusion of other HCS genes will improve cancer risk prediction [8,9,43]. In many studies, participants have volunteered for HCS testing [7] even when there are impediments such as financial cost involved [44], and it has been shown that receiving population-based testing results does not adversely affect participant's long-term psychological wellbeing or quality of life [7,40–42]. These participants have high rates of follow up with preventative care [7,40,44].

**Table 1.** Summary of select population-based genetic testing studies involving gynecological cancers.

| Reference | Population | Genetic Testing | Main Findings |
|---|---|---|---|
| Metcalfe 2010 [7]<br>Metcalfe 2013 [40] | AJ * women 25–80 y<br>Ontario, Canada (*n* = 2082) | Three *BRCA1/2* AJ<br>founder mutations | - High level of interest in testing among AJ women<br>- Many individuals with mutations would not have been eligible for testing under current guidelines<br>- ~1% of AJ women carry a *BRCA* mutation |
| Manchada 2015 [41]<br>Manchada 2020 [42] | Randomized controlled trial of AJ women/men (*n* = 1034) >18 y in Northern London | Three *BRCA1/2* AJ<br>founder mutations | - Population-based testing did not adversely affect long-term psychological wellbeing or quality of life compared to family-history based *BRCA* testing<br>- Population-based testing could identify up to 150% additional *BRCA* carriers |
| Narod 2021 [44]<br>The Screen Project | Canadians >18 y, open recruitment (*n* = 1269) | *BRCA1/2* mutations | - 2.4% carrier rate for a pathogenic *BRCA* mutation |
| Grzymski 2020 [8]<br>Healthy Nevada Project | Population-based cohort <18 y (*n* = 26,906) | *BRCA1/2, MLH1, MSH2, MSH6, PMS2* | - 1.33% carrier rate for pathogenic variants<br>- 90% of carriers had not been previously identified |
| Rowley 2019 [9]<br>Lifepool Australia | Women without cancer 50–74 y (*n* = 5908) | *BRCA1/2, PALB2, ATM, CDH1, PTEN, STK11, TP53, BRIP1, RAD51C/D* | - 0.64% of women carried a pathogenic variant<br>- Genetic testing was well accepted<br>- The majority of carriers would not have met existing family history testing eligibility |
| Hu 2021 [43] | Breast cancer (n = 32,247)<br>Controls (*n* = 32,544) | *ATM, BARD1, BRCA1/2, CDH1, CHEK2, NF1, PALB2, PTEN, RAD51C/D, TP53* | - 1.63% of controls had pathogenic variants |
| Gabai-kapara 2014 [45] | Population-based cohort of AJ men (*n* = 8195) | *BRCA1/2* | - 2.17% pathogenic variant prevalence |
| Dorling 2021 [39] | Breast cancer (*n* = 66,466)<br>Controls (*n* = 53,461) | 35 gene panel | - 2.0% of European controls had a pathogenic variant in breast-cancer associated gene |

\* AJ: Ashkenazi Jewish.

*1.4. Problem*

Review of the existing evidence confirms that expanded access to genetic testing for hereditary cancer predisposition has the potential to meaningfully impact cancer morbidity and mortality in Canada, and that new strategic directions in the model of hereditary cancer prevention are needed. This research summit proceedings reports on the expert discussion and conclusions.

**2. Methodology**

The Gynecologic Cancer Initiative (GCI) is a provincial network of scientists, physicians, researchers, and patient/family partners with a mission to accelerate transformative research on prevention, detection, treatment, and survivorship of gynecologic cancers and reduce the incidence, death, and suffering by 50% by 2034. The GCI hosted a multidisciplinary research summit in Vancouver, British Columbia (BC), Canada on 26 November 2021. The aim of the summit was to explore the potential of genetic testing in cancer prevention and address the gaps between the current state of evidence and the current

delivery of clinical care in the province of BC. International experts in gynecologic oncology, risk assessment epidemiology, genetics, and population health convened to review the evidence about inherited gynecologic cancer prevention and discuss the opportunities, barriers, and preferences for prevention at a population health scale.

The summit was organized around eight expert presentations focused on the key themes of "genetic and non-genetic factors in risk assessment and prevention", and "inverting the process—population-based genetic testing". In total, 44 attendees participated in the event, including physicians, researchers, patients, and community partners. Two patient partners were invited to attend and present their stories at the research summit. Each key theme was followed by a panel/group discussion to distill key points, benefits, and challenges. The goal of the meeting was to compile a list of priority items for the successful development of population-based testing and hereditary cancer prevention at a population, public health level. As a follow-up to the conference, attendees and presenters completed a survey about their conclusions and vision for the future of hereditary cancer prevention.

## 3. Results of Panel Discussion

*3.1. Preparing the Path from Patient-Driven Genetic Testing to Population-Based Genetic Testing. Why Do We Need to Wait for a Cancer to Happen to Identify People in Whom We Can Prevent Cancer?*

### 3.1.1. Paradigm Shift: Family History Based to Population-Based Testing

There is a need to "think big" about cancer and hereditary risk. Health systems and cancer genetics departments rely on family history-based eligibility criteria, but there is mounting evidence supporting a population-based approach to testing. Studies confirm that population-based *BRCA* testing would be cost-effective [46] and have a substantial impact on cancer morbidity and mortality. More than 50% of HCS carriers are not aware of a family history of cancer [42,47]. Organized processes offering testing to all women would seek to ensure equitable access to prevention for underserved populations. A population health view of hereditary cancer would translate to more equity of care, and importantly ensure that equity-deserving groups are included in research and development of new evidence about prevention and care [48].

### 3.1.2. Clinical Utility and Clinical Risk Management

While whole genome sequencing of every person in the population would provide an invaluable resource for the research community, from a practical standpoint, there is little immediate value in testing for disorders without clinical actionability. Reports of non-pathogenic variants or PRS do not yield a diagnosis but provide additional risk information. It will be important to ensure realistic expectation among both clinicians/patients about practical use of these tools. As knowledge about clinical efficacy advances, there is an argument to be made for pre-emptively testing variants of uncertain clinical utility with the hope that this may inform treatment decisions for patients in the future. The threshold of gene selection for a proposed population-based strategy will need to include disorders that are actionable now, but also allow for development of new evidence.

Successful implementation of new programs will require clear processes and professionals delivering results, as well as systems that support high-risk patients as they access preventative services over time. Organized patient navigation and provision of screening/surgery is highly variable in Canada, and patients consistently report difficulties in co-ordination of recommended care. Large-scale implementation of germline testing will require parallel development of hereditary cancer registries [49–52]. Registries can be run at low cost but deliver high value as they ensure high-risk individuals access effective prevention, evidence-based care, patient support, and access to research.

### 3.1.3. Psychological Impacts on Patients and Public Acceptability of Population-Based Testing

A key consideration in the success of a population-based genetic cancer testing program is whether it will be acceptable to the general population. This includes (1) will people be willing to undergo HCS testing, and (2) will they act on the results if they are found to be at-risk. Evidence from investigators and clinicians participating in this workshop indicate high public interest in testing for actionable genes and that a very small proportion of patients decline information about results [25]. High uptake of direct-to-consumer genetic testing demonstrates high general interest; however, products through private ancestry/health companies are neither clinical nor diagnostic. Pricing of private clinical testing at $200–400 US is expensive for most individuals [44,53].

Programs will require patient-centered processes considering the psychological impact of increased genetic risk. Public education reinforcing the value of prevention and ongoing individual support within hereditary cancer registries will mitigate these concerns. It has been demonstrated that testing of *BRCA1/2* and other cancer risk variants is broadly acceptable to population-based cohorts from Ashkenazi Jewish and general populations—there is appreciable interest in self-referral for genetic testing among low-risk women, with results indicating long-term psychologic distress was low and uptake of preventative interventions was high [7,23,40–42]. Genetics personnel to deliver results, explain concepts of risk, and provide continuity of care to high-risk patients will ensure the success of hereditary cancer registries.

### 3.2. Establishing the Social and Economic Context in Which Increased Genetic Testing Will Be Feasible and Acceptable. How Can the Canadian Model of Gynecologic Cancer Prevention Better Utilise Genetic and Genomics at a Population Level?

#### 3.2.1. Developing the Infrastructure to Support Increased Genetic Testing

Traditionally, genetic counselling was conducted via in-person sessions before and after genetic testing. This process was needed at a time when few genetic tests were available and risk assessment as well as clinical recommendations were based on expert pedigree analysis. As genetic testing has become more available and affordable and is increasingly required to guide cancer prevention or treatment, the former model of care may no longer address patient needs or permit timely access to care. More practical and effective models of genetic counselling will include group sessions, oncology clinic-based testing, counselling helplines and online interactive platforms [11,23,54,55]. These models have been shown to be satisfactory to patients and can deliver high quality information to more patients effectively at low cost [54,56,57]. The construction of a population-based testing program will require efficient models of care to deliver pretest education, communicate findings, and inform prevention or surveillance strategies.

To scale genetic testing on a population scale, requirements for additional infrastructure accommodating sample collection, personnel, space/equipment, data analysis/storage/linkage must be considered. These systems will need to be forward-thinking and adaptable: updating registries with current research knowledge and recall systems to reassess and recontact patients as new recommendations become available.

In the development of new clinical technical infrastructure, there is an opportunity to establish data harmonization standards that can be used widely as genetic testing becomes more prevalent [58,59]. Data governance could become a barrier to the collection of rich datasets because data sharing is often restricted by data localization requirements, consent, and strict data anonymization. There is a need for high-level data stewards to advise on data protection laws and establish frameworks for effective data sharing between hospitals and health authorities. Patients will have a powerful voice in the development of these processes and health policies and opportunities to participate in research must be equitable.

### 3.2.2. Ensuring Equitable Access to Genetic Testing

For a patient to access hereditary cancer prevention in the current model of care, several steps are required; detailed knowledge of one's family history of cancer, an understanding of health literacy and genetics, access to a referring physician, and the ability to travel to a genetics clinic or engage via telehealth. The complexity of this process has meant that educated individuals of white European ancestry or high socioeconomic status are over-represented in genetics programs across Canada. People of color, rural/remote communities, and Indigenous populations do not receive the benefit of prevention in this model. Indigenous communities face unique health challenges, inequities, and access to health care and typically have poorer health outcomes than non-Indigenous groups. Collection of a family history itself for Indigenous families is complex and can be traumatizing. A recent study on ethnic distribution of hereditary cascade genetic testing in British Columbia highlighted a significant underrepresentation of Indigenous individuals and recognized the need for culturally safe alternatives to outreach and service promotion [19]. The ongoing Silent Genome project whose goal is to securely sequence and database the genomes of approximately 1500 Indigenous, First Nations, and Métis Canadians [60], aims to address the inequities in access to genetic research and testing in Indigenous peoples in Canada. Amongst the many reasons a health system would implement population-based testing, the need for equity in genetics care and prevention is clearly the most compelling.

### 3.2.3. Economic Feasibility and Sustainability

In an era of an aging demographic, increasingly complex cancer treatments, longer survivorship, and the debut of costly targeted therapeutics, a cancer care system that focuses on treatment over prevention will not be sustainable. Management of chronic diseases including cancer are a substantial proportion of health care budgets [61]. Hereditary cancer prevention has the potential to decrease spending on cancer treatment but must provide a value proposition. Population-based testing for *BRCA1/2* in women $\geq$30 years has been estimated in UK and US health systems, and this testing has the potential to be cost-saving (incremental cost-effectiveness ratio (ICER) per quality-adjusted life year (QALY) of $-5639$/QALY and $-4018$/QALY in the UK and USA, respectively [46]. The cost incurred with genetic testing is far outweighed by the cost avoided in treating ovarian cancer. There is now an urgent need for this work in a Canadian model. Although it will likely prove as well to be cost-saving in the context of this health care system, further evidence from clinical and implementation science research is required. Considering that one-fifth of ovarian cancer patients [62] and 9% of breast cancer patients [63] carry at least one HCS pathogenic variant, a population-based genetic testing strategy has the potential to prevent over 80 cases of ovarian cancer in British Columbia and over 750 in Canada annually. Data suggest that expanding this to a panel of breast and ovarian cancer genes would also be cost-effective for the health system [64].

Ultimately, prevention of chronic diseases and improved population health will be the key to building a more financially sustainable healthcare system.

### 3.3. Actionable Steps towards Adopting Population-Based Genetic Cancer Testing

Implementation of a population-based genetic testing program for cancer prevention is a long-term goal and will need to be accomplished in iterative stages along with the development of appropriate infrastructure. Specific areas for actionability that can improve the more immediate prevention of hereditary cancer are illustrated in Figure 1.

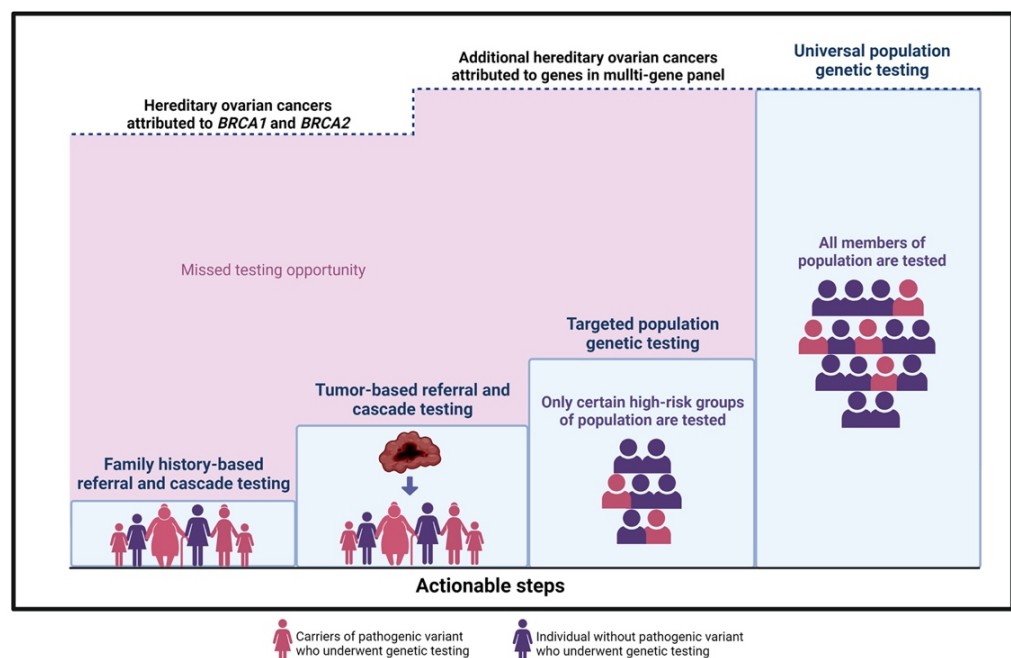

**Figure 1.** Actionable steps towards hereditary ovarian cancer prevention. The upper dashed line represents the ~20% of hereditary ovarian cancer cases that are attributed to *BRCA1* and *BRCA2* mutations.

### 3.4. Recommended First Steps

(1) **Family history-based referral and cascade testing:** Strategies to increase detection of high-risk individuals would improve the uptake of testing in relatives of HCS carriers. i.e., a sister of a confirmed *BRCA* carrier is at 50% risk of having the familial mutation. In the current model, an index patient is responsible for communication with family members. This strategy has low uptake, with fewer than 50% of high-risk relatives accessing care. Direct relative communication via health professionals increases the uptake of cascade testing [65]. Studies in a UK model evaluating the effectiveness of registries in the facilitation of family contact confirm that registries provide long-term follow-up and proactive genetic counselling to relatives at risk, potentially an effective initiative in the Canadian context. The shift towards a formal provider-initiated, registry-based cascade testing is optimal, but important considerations must balance the concept of "duty to warn" with individual confidentiality and privacy [66].

(2) **Tumor-based referral and cascade testing:** A tumor-first approach can be employed where samples from surgical specimens are sequenced, triggering a stepwise process whereby tumor genomic results are reported routinely and then patients with possible HCS are counselled regarding germline testing. This strategy would be delivered universally, without barriers, and would allow more equitable care across race or socioeconomic status. Patients may directly benefit from information about potential targeted therapies [67,68]. This approach could also offer the option of improving family risk assessment via tumor sequencing from deceased patients for the benefit of relatives and could be delivered in a way to respect individual preferences of patients about germline testing.

(3) **Targeted population-based genetic testing:** A strategic step toward increased detection of HCS carriers in a population would expand testing to specific groups or population subsets with higher mutation prevalence; this approach is now standard care for all ovarian cancer patients and has been successfully implemented across Canada. By removing the requirement of a family cancer history for testing eligibility, the process of offering testing to all patients in a category, i.e., breast cancer [69] or pancreatic cancer [70], irrespective of family history could meaningfully reduce barri-

ers to care [71]. Testing all women with breast cancer for HCS genes has been shown to be cost-effective for UK and USA health systems [72]. Testing all individuals with Ashkenazi Jewish descent will identify *BRCA* mutations in 2–3% of unselected cases and has been proven (1) acceptable to patients, (2) clinically effective, and (3) cost saving for health systems [42,73,74]. A logical first step in any planned expansion of prevention would begin with publicly funded testing for Ashkenazi Jewish Canadians [75,76] and broader eligibility testing criteria for patients with specific cancers. Populations experiencing reduced access, care, and family history assessment, i.e., Indigenous communities, would be prioritized. These higher risk groups could be approached first for population-based testing initiatives [77–81].

### 3.5. Cancer Prevention Summit Participant Survey Results

Of 44 attendees, 21 completed post conference surveys about the future of hereditary cancer prevention and priorities in the development of population-based testing strategies. Summit participants responded to Likert-scale questions about conclusions from the meeting and key research priorities (Figures 2 and 3).

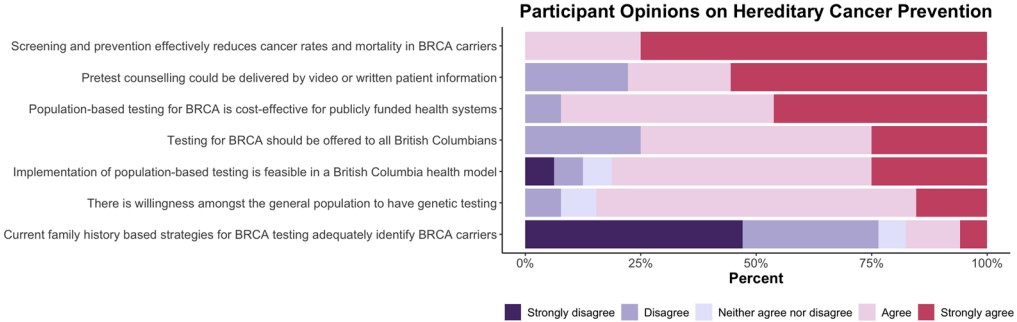

**Figure 2.** Research summit participants were asked to rate statements about hereditary cancer prevention using the following scale: strongly disagree, disagree, neither agree nor disagree, agree, or strongly agree.

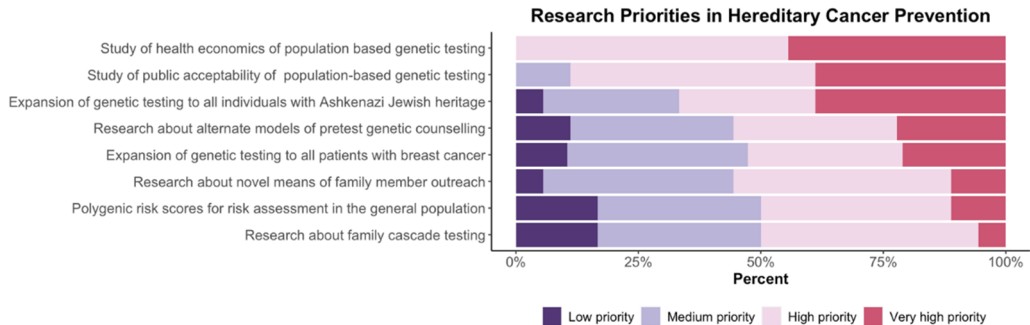

**Figure 3.** Research summit participants were asked to assign a priority rating to hereditary cancer prevention action items using the following scale: low priority, medium priority, high priority, or very high priority.

As noted in Figure 2, there is strong support from participants for screening and prevention through population-based testing programs. Respondents reported consistent agreement that preventative interventions effectively reduce cancer rates and morbidity and that population-based testing can be feasible and acceptable to the public. It was noted by more than 75% of participants that current family history-based strategies were ineffective. Figure 3 illustrates participants' responses about the recommended priority research areas and that key areas of focus are (a) health economics assessment and (b) evaluation of public acceptability.

## 4. Discussion

This report summarizes the findings of the British Columbia Gynecologic Cancer Initiative conference addressing the potential of population-based testing for hereditary cancer prevention. The team concludes the following key points:

(1) Implementation of new technologies and prevention strategies must be developed in a way that is equitable to all individuals in a population, regardless of ethnicity, socio-economic status, or geography. Every effort to remove elements of institutional racism in the delivery of new systems is essential. In communities where family-history-based requirements for testing are a barrier, those populations, specifically Indigenous peoples, should be provided first offer to engage in the development of broad population-based testing initiatives.

(2) Population-based testing for genetic risk should become the standard of care for effective cancer prevention and could be cost-effective for the long-term sustainability of health systems. Before population-based testing is implemented, incremental improvements in the identification of high-risk individuals will deliver more effective cancer prevention. This will include testing for all individuals with Ashkenazi Jewish heritage, relatives of women with ovarian cancer, and expanded access to testing for patients with breast, prostate, and pancreatic cancer.

(3) Hereditary cancer registries providing high-risk patients with supportive navigation of preventative screening/surgery and access to research will be a key element in effective prevention programs.

(4) Successful population-based testing strategies will require infrastructure, well-developed public education models and must be guided by public preferences.

(5) There is need for improved patient access and more efficient delivery of germline testing within current hereditary cancer systems. This will require broader use of group pre-test counselling, testing embedded in cancer clinics, and use of digital patient-facing education and counselling tools.

(6) Cascade testing of family members is critical for success of prevention programs. Strategies to improve rates of testing in high-risk family members will include facilitated family communication, use of digital education/outreach tools, and broader public education.

(7) Polygenic risk scores and epidemiological risk assessment models have value in the delivery of personalized preventative interventions. These models may be implemented in parallel with germline testing at both a population level and in the context of proven *BRCA* mutations. This will become feasible as better validation data emerge and implementation studies follow.

## 5. Conclusions

This multi-disciplinary international workshop concludes that a broader, population-based testing strategy for hereditary cancer prevention could be a key component of successful chronic disease prevention in Canada. The recommended priority research areas are (a) Canadian health economics assessment and (b) Canadian evaluation of public acceptability.

The immediate priority health policy recommendations are (1) to expand genetic testing to high-risk groups, specifically individuals of Ashkenazi Jewish ancestry, and (2) to ensure that groups experiencing the most reduced access within the current model (i.e., Indigenous peoples) should be provided with the offer of first priority in the development of population-based testing initiatives.

**Author Contributions:** Conceptualization, G.S., D.H. and L.D.; writing—original draft preparation, L.C.T., A.Z., S.L. and L.D.; writing—review and editing, L.C.T., A.Z., S.L., M.W., J.S.K., G.E.H., B.K., K.A.S., S.J.P., A.T., T.D., K.M., N.P., W.D.F., R.M., D.H., G.S., J.S. and L.D.; visualization, L.C.T. and A.Z.; supervision, L.D.; project administration, S.L. and M.W.; funding acquisition, L.D. All authors have read and agreed to the published version of the manuscript.

**Funding:** We are grateful to the Homi Maneck Italia Foundation supporting this research summit.

**Acknowledgments:** We thank all speakers and attendees of the Gynecologic Cancer Initiative prevention summit for their participation. We are especially grateful to Rose Lee and Shiraz Italia for sharing their personal stories and their dedication to prevention.

**Conflicts of Interest:** The authors declare no conflict of interest.

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
