# Peer review of "Gynecologic Cancer Risk and Genetics: Informing an Ideal Model of Gynecologic Cancer Prevention"

_curroncol, doi:10.3390/curroncol29070368_

Round 1

Reviewer 1 Report

This work presents the proceedings of an “expert summit” engaging a small, albeit qualified, group of participants. It draws an “ideal” model based on literature evidence and a little internal little survey.

The results of the summit offer suggestions and possible perspectives about an “ideal model”: the review can therefore only offer general consideration on the same general level. However, every public health program must face a precise target, feasibility criteria, cost effectiveness evaluation and practical solutions deriving from the concrete local scenario and health care organization, not only from the literature evidence.

The ”gold standard” identified by this ideal model is to offer a genetic test to all women (without any age restriction?). Several weaknesses and unspecified aspects of this model seem to emerge from the paper:

  • How many women do you esteem to involve?
  • The program doesn’t take into consideration the opportunity and the availability of tools for a previous evaluation of cancer onset lifetime risk, to avoid unnecessary tests in women that are not at risk, needless costs and burden on the genetic units.
  • The prevention program appears to be based only on the offer of a genetic test, with poorly defined pre-test and post-test global assistance. This is a serious flaw: genetic test results give not only negative/positive results. What is the path for variants of uncertain significance?
  • The ideal strategy appears to be focused mainly on RRSO as a prevention strategy for ovarian cancer: what about the entire path for breast cancer prevention (surgical, medical) that a universal genetic test need? What about the path for women with negative BRCA test but high lifetime risk? Nothing is said about a possible collaboration with cancer screening programs to draw prevention strategies for these women.
  • Health economics assessment investigations are entrusted to subsequent phases of the study, but some premises anyway provide an incomplete overview of the topic. Cancer prevention is undoubtedly cost-effective than cancer therapy, but while the costs are immediate, the “earnings” are subordinated to the (long) time and to the effective coverage of the program, not always so easy, as the screening programs experiences have demonstrated. Obviously, to be “universal”, the entire path must be free, including all supporting organizational structures as test offer management, registries of participating women, cancer registries (which do not have to be “hereditary” but “general” to ensure the evaluation of outcomes. They are anything but “low-cost”)
  • An “expanded” access for prostate and pancreatic cancer patients is mentioned without any consideration about specific paths, efficacy/cost-effectiveness/feasibility deepening.

This ideal model offers interesting ideas, but it appears too “general” and still premature to configure a feasible project. For these reasons, it doesn’t still appear ready for publication.

Author Response

Thank you to reviewers and Current Oncology staff.

Reviewer #1 has several high-level observations about the manuscript that make excellent points with which we agree. There is a comment that “The prevention program appears to be based only on the offer

of a genetic test, with poorly defined pre-test and post-test global assistance”. If it is not clear that these strategies would be directly embedded in the broad-based provincial hereditary cancer program (HCP) in BC, then we hope that we have added clarity to the manuscript. In both the participant survey and the
discussion, we have carefully included considerations about the needed models of pretest counselling and care delivery. The reviewer is correct in stating that posttest global assistance is essential for the successful delivery of preventative care, and we have commented in the text and conclusion about the importance of hereditary registries and patient navigation and support. We fully agree that the implications of health economic assessment are not just short-term considerations and must be offset by the costs of cancer therapy. 

Lesa Dawson MD FRCSC

Reviewer 2 Report

This is a generally well-written paper on an important topic (expanding access and use of genetic testing in preventive oncology). It's exciting that there is enough momentum for an entire conference on this topic in BC and I think that the information presented in the manuscript is worthy of reporting in the published literature. Below are my suggestions for improving the paper before publication:

-Define "population-based" early in the manuscript and be VERY clear when you are using different conceptualizations of "population-based". The manuscript often conflates population based (entire adult population) and testing of much more target populations. This nuance is especially important when discussion testing people w/ and w/out cancer, which has different risks and benefits.

-I would suggest pulling back to focus (and excitement) about polygenic risk scores throughout, but especially in the introduction. There is limited evidence supporting their clinical utility and, more importantly, the results of the primary data collected from your meeting attendees doesn't seem to warrant the amount of attention paid to them in manuscript. 

-Along these lines, significantly more should be done to interpret and comment on the survey results shown in Figure 2 and align the survey results with the content and recommendations made in the paper.

-Greater nuance with language (especially causal language) about representation in currently tested populations and reasons for under-representation. We do not that the patterns we see are all because of "barriers", which you need to better define and reference, or some combination of "barriers" and preferences. A lot of attention is given to the need to address barriers, especially in Indigenous people, based on what seems like one study. The current manuscript also makes it seems like "barriers" disappear w/ population-based testing, which is not true. Some barriers are minimized (complex risk assessment and referral criteria), but new/other barriers emerge. 

-Some mention of the inherent conflict of implementing the process shown in Figure 1 needs to be given: some people may not agree with an approach that advocates for improving the current system, but then ultimately moves away from that system to another system. Why both with the incremental approach?

-There are lots of references included, but also many statements in the paper that seem like then need references supporting the claims being made, but don't have them. 

  -Table 1: Are all of these trials? How are you defining a trial? The population column does not consistently list the population targeted for test. It's a combination of study design, population, and recruitment/outreach strategy.  Genetic testing column is inconsistent in the way it talks about testing. List the genes and variants (some, all).

-Figure 2,3. Could you flip the order of the bars in one of the figures so they are aligned between the two? Right now the top one shows those with most agreement first/highest and the bottom one shows those with the most priority last/lowest. 

-On page 6 I am not sure what you mean by "preemptively testing variants of uncertain clinic utility"? Identifying and reporting variants of uncertain significance (VUS)? Or looking for pathogenic/likely path variants in genes where the relationship with cancer risk is unknown?  

Author Response

Thank you to reviewers and Current Oncology staff.

Reviewer #2 has a series of thoughtful questions and suggestions that we have addressed as follows:
“This is a generally well-written paper on an important topic (expanding access and use of genetic testing
in preventive oncology). It's exciting that there is enough momentum for an entire conference on this topic
in BC and I think that the information presented in the manuscript is worthy of reporting in the published
literature. Below are my suggestions for improving the paper before publication”:
1. Define "population-based" early in the manuscript and be VERY clear when you are using
different conceptualizations of "population-based". The manuscript often conflates population
based (entire adult population) and testing of much more target populations. This nuance is
especially important when discussion testing people w/ and w/out cancer, which has different
risks and benefits.
Response: We have entered a clear definition on page 2 of the manuscript

  1. I would suggest pulling back to focus (and excitement) about polygenic risk scores throughout,
    but especially in the introduction. There is limited evidence supporting their clinical utility and,
    more importantly, the results of the primary data collected from your meeting attendees doesn't
    seem to warrant the amount of attention paid to them in manuscript.
    Response: We would agree and have, therefore, redacted the paragraphs detailing the potential
    for PRS.
    3. Along these lines, significantly more should be done to interpret and comment on the survey
    results shown in Figure 2 and align the survey results with the content and recommendations
    made in the paper.
    Response: On page 10, we have inserted a paragraph setting the context for the inclusion of the
    survey results.
    4. Greater nuance with language (especially causal language) about representation in currently
    tested populations and reasons for under-representation. We do not know that the patterns we see
    are all because of "barriers", which you need to better define and reference, or some combination
    of "barriers" and preferences. A lot of attention is given to the need to address barriers,
    especially in Indigenous people, based on what seems like one study. The current manuscript also
    makes it seems like "barriers" disappear w/ population-based testing, which is not true. Some
    barriers are minimized (complex risk assessment and referral criteria), but new/other barriers
    emerge.
    Response: We accept and agree with the reviewers comments. We have removed the word
    “barrier” where appropriate and referred to concept of equitable access and preferences which is
    more accurate.
    5. Some mention of the inherent conflict of implementing the process shown in Figure 1 needs to be
    given: some people may not agree with an approach that advocates for improving the current
    approach that advocates for improving the current system, but then ultimately moves away from
    that system to another system. Why both with the incremental approach?
    Response: Thank you for this valuable comment. This was a point of great discussion. It is
    believed that broad immediate implementation of population-based testing is not expected to be
    possible until further work about public acceptability and public acceptance have been completed.
    The development of such a program will take several years at least.Our team is undertaking this
    work at present. As such the participants felt that expansion of testing in other populations i.e.,
    individuals with Ashkenazi Jewish ancestry, cancer patients, underserved communities could
    make immediate impact. Current evidence available today supports broader implementation of
    genetic testing in these groups.
    6. “There are lots of references included, but also many statements in the paper that seem like the
    need references supporting the claims being made, but don't have them.”
    Response: We have reduced the number of references significantly and used the format for these
    as outlined in the MDPI Instructions for Authors.
    7. Table 1: Are all of these trials? How are you defining a trial? The population column does not
    consistently list the population targeted for test. It's a combination of study design, population,

and recruitment/outreach strategy. Genetic testing column is inconsistent in the way it talks about
testing. List the genes and variants (some, all).
Response: Most of these are not trials and would be more correctly referred to as “studies”. As
such, the Table has been revised.
8. Figure 2,3. Could you flip the order of the bars in one of the figures so they are aligned between
the two? Right now, the top one shows those with most agreement first/highest and the bottom
one shows those with the most priority last/lowest.
Response: This has been done.
9. On page 6 I am not sure what you mean by "preemptively testing variants of uncertain clinic
utility"? Identifying and reporting variants of uncertain significance (VUS)? Or looking for
pathogenic/likely path variants in genes where the relationship with cancer risk is unknown?
Response:
This point was discussed in detail, and the attendees agreed that broad reporting of VUS or
unknown genes should not be undertaken in a new population-based model.

Lesa Dawson MD FRCSC

Round 2

Reviewer 1 Report

This work, also in its revised form, stresses the need to more effectively and quickly identify women at hereditary risk for gynecological cancer, but leaves intact several perplexities previously expressed:

  • About 20% of ovarian cancers and <5% of breast cancer are attributable to a heritable risk (mainly based, at the present time, on BRCA 1-2 mutations). It means that over 80% of women will receive a genetic test without any effective risk. For this reason, the choice of a direct offer of a genetic test without any family anamnestic referral for the whole population (the project doesn’t’ have any age restriction) appears not efficient.
  • Most of the other aspects seem to be underestimated: nothing is said about an esteem of women involved, pre/post test concrete support and, above all, about managing women with tests with uncertain meaning. This is a serious weakness. Implementing a hereditary cancer registry is by no means a "low-cost" operation, as the experience of cancer registries teaches. It could be better to enrich a population cancer registry with information on genetic risk. A “general” population cancer registry is required to assess the real impact of the program.
  • If surgery represents the best prevention choice for ovarian cancer in high-risk women, this is not the only chance for breast cancer. Nothing is said about practical solutions for the management of breast cancer risk in women with positive genetic test: this represents a serious problem with enormous ethical implications that needs to be discussed and solved.

Practically all the problems reported during the first review remains on the table. This is a good preparatory study with an exhaustive literature examination, but not a "model", since it doesn’t consider important aspects of a concrete health care project. 

Author Response

This work, also in its revised form, stresses the need to more effectively and quickly identify women at hereditary risk for gynecological cancer, but leaves intact several perplexities previously expressed:
About 20% of ovarian cancers and <5% of breast cancer are attributable to a heritable risk (mainly based, at the present time, on BRCA 1-2 mutations). It means that over 80% of women will receive a genetic test without any effective risk. For this reason, the choice of a direct offer of a genetic test
without any family anamnestic referral for the whole population (the project doesn’t’ have any age restriction) appears not efficient.

We agree with Reviewer 1 that that majority of women who would receive a genetic test with universal testing would not be positive for a cancer-associated mutation. Many women, however, are diagnosed with cancer, have no family history, and are later found to have a known hereditary mutation. In this
manuscript we assert that these women represent an important population for whom we could potentially prevent a cancer. More that 50% of BRCA carriers will not report a family history, meaning that FHx criteria are not adequate. Under the current clinical model, this prevention opportunity is missed. We are proposing that an “ideal” model of cancer prevention would serve all people and reduce barriers. Given the dramatic decrease in testing costs, a change in strategy is possible.

Most of the other aspects seem to be underestimated: nothing is said about an esteem of women involved, pre/post test concrete support and, above all, about managing women with tests with uncertain meaning. This is a serious weakness. Implementing a hereditary cancer registry is by no means a "low-cost" operation, as the experience of cancer registries teaches. It could be better to enrich a
population cancer registry with information on genetic risk. A “general” population cancer registry is required to assess the real impact of the program.

We commend the reviewer’s comments about the complexity of these topics. The goal of this manuscript is to reflect the depth of the discussions at this conference and explore the practicalities of implementation. We respectfully disagree that the need for patient-centered pre/post-test care has not
been acknowledged by this team. It is clear that current system of care is not meeting the needs of proven mutation carriers, and this initiative is focused on a pathway to better serve these patients and families. Development of future effective registries will require thoughtful implementation and collectively this team is committed to contributing to these discussions and evidence.

If surgery represents the best prevention choice for ovarian cancer in high-risk women, this is not the only chance for breast cancer. Nothing is said about practical solutions for the management of breast cancer risk in women with positive genetic test: this represents a serious problem with enormous ethical
implications that needs to be discussed and solved.

The evidence for both breast MRI or mastectomy in BRCA carriers is clear, either option improves outcomes, and clinical teams support women in the personalized decision making.

Practically all the problems reported during the first review remains on the table. This is a good preparatory study with an exhaustive literature examination, but not a "model", since it doesn’t consider important aspects of a concrete health care project.

We thank the reviewer for these comments and look forward to further work in this area.